# Implementation, feasibility, and acceptability of 99DOTS-based supervision of treatment for drug-susceptible TB in Uganda

Noah Kiwanuka[1]☯, Alex Kityamuwesi[2]☯, Rebecca Crowder[3], Kevin Guzman[4], Christopher A. Berger[3], Maureen Lamunu[2], Catherine Namale[2], Lynn Kunihira Tinka[2], Agnes Sanyu Nakate[2], Joseph Ggita[2], Patricia Turimumahoro[2], Diana Babirye[2], Denis Oyuku[2], Devika Patel[5], Amanda Sammann[5], Stavia Turyahabwe[6], David W. Dowdy[7], Achilles Katamba[2,8‡], Adithya Cattamanchi[2,3,9‡]*

1 Department of Epidemiology & Biostatistics, School of Public Health, College of Health Sciences, Makerere University, Kampala, Uganda, 2 Walimu, Uganda Tuberculosis Implementation Research Consortium, Kampala, Uganda, 3 Center for Tuberculosis and Division of Pulmonary and Critical Care Medicine, San Francisco General Hospital, University of California San Francisco, San Francisco, California, United States of America, 4 Department of Medicine, University of California San Francisco, San Francisco, California, United States, 5 Department of Surgery, San Francisco General Hospital, University of California San Francisco, San Francisco, California, United States of America, 6 Uganda National Tuberculosis and Leprosy Programme, Ministry of Health, Kampala, Uganda, 7 Department of Epidemiology, Johns Hopkins Bloomberg School of Public Health, Baltimore, Maryland, United States of America, 8 Department of Medicine, School of Medicine, College of Health Sciences, Makerere University, Kampala, Uganda, 9 Division of Pulmonary Diseases and Critical Care Medicine, University of California Irvine, Irvine, California, United States of America

☯ These authors contributed equally to this work.
‡ AK and AC contributed equally to this work as senior authors.
* Adithya.Cattamanchi@uci.edu

**Data Availability Statement:** All relevant data are within the manuscript and its Supporting Information files.

## Abstract

99DOTS is a low-cost digital adherence technology that allows people with tuberculosis (TB) to self-report treatment adherence. There are limited data on its implementation, feasibility, and acceptability from sub-Saharan Africa. We conducted a longitudinal analysis and cross-sectional surveys nested within a stepped-wedge randomized trial at 18 health facilities in Uganda between December 2018 and January 2020. The longitudinal analysis assessed implementation of key components of a 99DOTS-based intervention, including self-reporting of TB medication adherence via toll-free phone calls, automated text message reminders and support actions by health workers monitoring adherence data. Cross-sectional surveys administered to a subset of people with TB and health workers assessed 99DOTS feasibility and acceptability. Composite scores for capability, opportunity, and motivation to use 99DOTS were estimated as mean Likert scale responses. Among 462 people with pulmonary TB enrolled on 99DOTS, median adherence was 58.4% (inter-quartile range [IQR] 38.7–75.6) as confirmed by self-reporting dosing via phone calls and 99.4% (IQR 96.4–100) when also including doses confirmed by health workers. Phone call-confirmed adherence declined over the treatment period and was lower among people with HIV (median 50.6% vs. 63.7%, p<0.001). People with TB received SMS dosing reminders on 90.5% of treatment days. Health worker support actions were documented for 261/409 (63.8%) people with TB who missed >3 consecutive doses. Surveys were completed by 83

**Funding:** This project was supported by the Stop TB Partnership's TB REACH initiative and was funded by the Government of Canada, the Bill & Melinda Gates Foundation, and the United States Agency for International Development (AC and AKa), as well as the UCSF Nina Ireland Program for Lung Health (CAB). The funders had no role in study design, data collection and analysis, decision to publish, or preparation of the manuscript. AKi, RC, CAB, ML, LKT, JG, PT, DP, AKa, AC received salary support from the TB REACH grant and CAB received salary support form the UCSF Nina Ireland Program for Lung Health grant.

**Competing interests:** The authors have declared that no competing interests exist.

people with TB and 22 health workers. Composite scores for capability, opportunity, and motivation were high; among people with TB, composite scores did not differ by gender or HIV status. Barriers to using 99DOTS included technical issues (phone access, charging, and network connection) and concerns regarding disclosure. 99DOTS was feasible to implement and highly acceptable to people with TB and their health workers. National TB Programs should offer 99DOTS as an option for TB treatment supervision.

## Author summary

This is the first study from sub-Saharan Africa reporting both implementation metrics and implementation feedback on 99DOTS, a low-cost DAT which is already widely used in India. The implementation assessment was nested within a pragmatic implementation trial of 99DOTS at 18 health facilities in Uganda with National Tuberculosis and Leprosy Program (NTLP)-affiliated TB treatment units. Using process metric data from the trial and theory-informed surveys to subsets of people with TB and health workers, the study demonstrated that 99DOTS is feasible to implement with high fidelity and more acceptable to both people with TB and health workers than previously reported in the literature. We found that 99DOTS is feasible to implement as an alternative method of TB treatment supervision, and highly acceptable to both people with TB and health workers in Uganda, supporting its further scale-up as an option for TB treatment supervision. The findings highlight the importance of adapting and contextualizing DATs prior to implementation, with specific attention to modifications that reduce stigma and enhance education/motivation as well as connection between people with TB and their health workers.

## Introduction

Poor and variable treatment completion persists as a key barrier to ending tuberculosis (TB) globally [1]. Since the scale-up of directly observed therapy (DOT) in the 1990s, the experience of TB treatment has changed little for people with TB. Recently, digital adherence technologies (DATs) have emerged as an alternative to DOT. DATs enable people with TB to take medicines on their own while facilitating remote monitoring and support of medication adherence by health workers. Although DATs are increasingly being deployed by National TB Programs, there are limited data on fidelity of implementation, acceptability, and feasibility, particularly in sub-Saharan Africa.

99DOTS is a low-cost DAT developed specifically for supporting TB treatment adherence. TB medication blister packs are packed in custom sleeves that reveal toll-free numbers (TFNs) when pills are removed. People with TB call the toll-free number to confirm medication dosing, and health workers can monitor adherence through a dashboard accessible via a smartphone or desktop computer [2]. 99DOTS has been scaled-up across India despite limited data about its effectiveness in supporting treatment adherence and completion. Studies have also raised implementation concerns [3–5] and there has been considerable debate as to whether DATs improve the experience of TB treatment for both people with TB and health workers [6].

Previously, we reported the first randomized trial of 99DOTS-based TB treatment supervision. The trial found that treatment completion did not improve overall but did improve among the 52% of people with TB enrolled on 99DOTS during the intervention period. Here,

we report on the implementation of key components of the 99DOTS-based intervention, as well as its feasibility and acceptability to people with TB and health workers.

## Results

During the study period, 463 people with pulmonary TB were enrolled on 99DOTS within the first month of TB treatment across 18 participating health facilities. One person with TB could not be matched to the TB register and was excluded. Of the remaining 462 participants, 296 (64%) were male, median age was 36 (IQR 28–48), 276 (60%) had bacteriologically confirmed TB, 422 (91%) had not been previously treated for TB, and 191 (41%) were living with HIV.

### Fidelity

**Calling TFNs to report dosing.** The median proportion of expected doses self-reported as taken by phone calls to 99DOTS was 58.4% (IQR 38.7–75.6) and fell from 71.4% (IQR 42.9–88.9, n = 456) during the first month of treatment to 46.4% (IQR 21.4–75.0 n = 412) during the 6th month of treatment (chi-squared test for trend: p<0.001, **Fig 1**). When doses confirmed by health workers were included, the median proportion of expected doses reported as taken was high overall (99.4%, IQR 96.4–100) and during each month of treatment (range: 96.4–100%) (**Fig 1**). Engagement with 99DOTS (*i.e.*, proportion of expected doses reported as taken

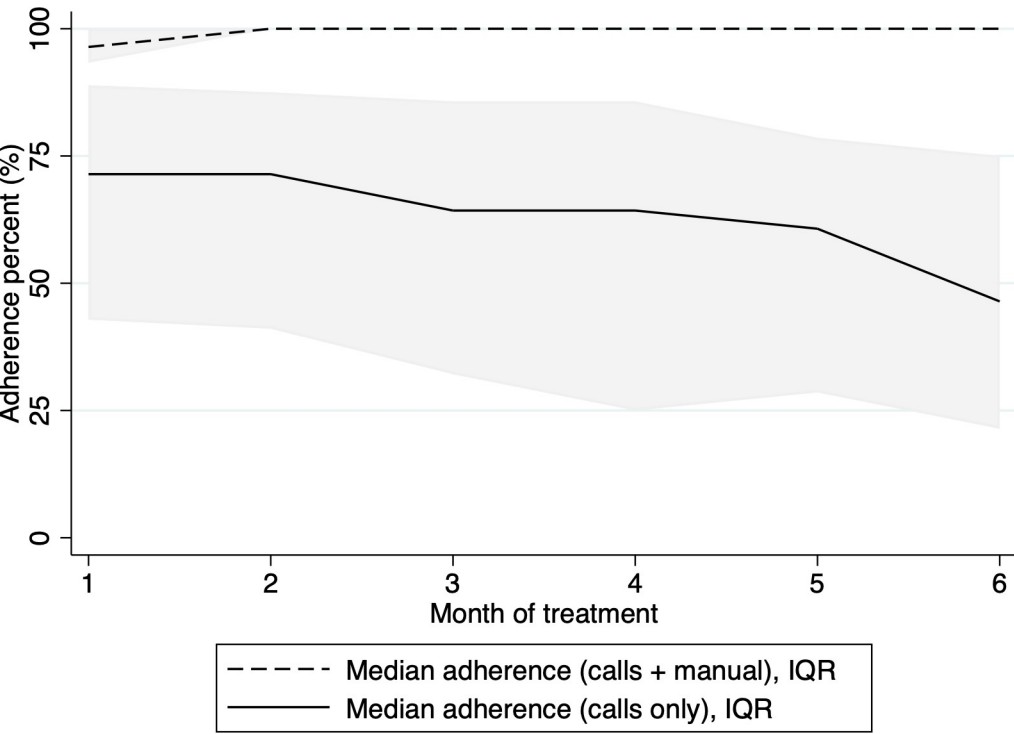

**Fig 1. Adherence by month of TB treatment, reported by calls to 99DOTS (solid line) or a combination of calls and doses added manually by health workers (dashed line).** Adherence over the course of treatment, measured during each 28-day month of treatment. The solid line indicates the proportion of doses confirmed by phone call. The dashed line indicates the proportion of doses confirmed either by phone call or by health workers (ascertained by self-report or phone calls to people with TB). The gray shading around each line indicates the inter-quartile range of adherence during each month of treatment. Median adherence recorded by phone calls was 71.4%, 71.4%, 64.3%, 64.3%, 60.7%, and 46.4% over months 1–6 of treatment. Median adherence after incorporating manual doses was 96.4% during the first month of treatment and 100% for months 2–6. IQR: Interquartile range; TB: Tuberculosis.

**Table 1. Fidelity to components of the 99DOTS-based intervention.**

| Daily short message service (SMS) dosing reminders | |
|---|---|
| Proportion of daily SMS sent to people with TB | 58,831 / 65,045 (90.5%) |
| Proportion of daily SMS received by people with TB | 58,547 / 58,831 (99.5%) |
| **Health worker support actions** | |
| Phone calls | |
| Number of supportive calls to people with TB | 980 |
| Proportion of people with TB who received a support call | 258 / 462 (55.8%) |
| Median number of calls per person receiving a call (IQR) | 2 (1–4) |
| Home visits | |
| Number of home visits to people with TB | 157 |
| Proportion of people with TB who received a home visit | 91 / 462 (19.7%) |
| Median number of visits per person receiving a home visit (IQR) | 1 (1–2) |
| Facility visits | |
| Number of supportive facility visits | 219 |
| Proportion of people with TB who came for a supportive facility visit | 103 / 462 (22.3%) |
| Median number of visits per person logging a facility visit (IQR) | 1 (1–3) |

IQR: Inter-quartile range

by people with TB calling TFNs) was similar among men and women (median 57.1% vs. 61.9%, p = 0.37) and by age quartile (**S1 Table**). However, people living with HIV reported fewer doses by calling TFNs than people living without HIV (median 50.6% vs. 63.7%, p<0.001); people living with HIV also had a greater decline in engagement over the course of treatment (p<0.001) (**S1 Table**, **S1 Fig**).

**Daily SMS reminders.** Of 65,045 eligible days, SMS messages were sent on 58,831 (90.5%), and 58,547 (99.5%) were successfully received on handsets belonging to people with TB (**Table 1**). The main reason for SMS sending failure was service outage, recorded on 29/357 (8.1%) days. Reasons for unsuccessful delivery of SMS to people with TB included incorrect phone numbers or phones being switched off during repeated SMS delivery attempts.

**Support actions by health workers.** Health workers were enabled to document support actions in the 99DOTS app in January 2019. Health workers recorded 980 phone calls made to 258 people with TB, 157 home visits to 91 people with TB and 219 extra facility visits made by 103 people with TB. Of 409 (88.5%) people with TB who did not call TFNs to report >3 consecutive doses at least once, 261 (63.8%) were documented to have received at least one type of support action (**Table 1**).

## Acceptability and feasibility

Of 86 people with TB enrolled on 99DOTS selected to complete the acceptability and feasibility survey, three were excluded for incomplete responses. Survey participants were similar to the remaining population with respect to demographic and clinical characteristics (**S2 Table**). Among survey participants, 33% had completed education past the equivalent of secondary school in the United States; 90% were employed; and the median household size was 5 persons (**Table 2A**). Access to a phone was difficult for some, with 14% sharing a phone, 18% changing their Subscriber Identity Module (SIM) card in the last year, and 71% not having consistent access to prepaid mobile airtime. Access to airtime was not needed to make calls to 99DOTS.

People with TB reported several methods for remembering to take their medication: SMS reminder from 99DOTS (n = 45, 54%), setting an alarm (n = 25, 30%), family or friends reminding them (n = 23, 28%), taking it in the morning or reminding themselves (n = 14, 17%), and hearing a radio program or call to prayer (n = 6, 7%). The majority (n = 47, 57%)

**Table 2. Self-reported characteristics of people with TB and health workers using 99DOTS.**

| A. People with TB surveyed (N = 83) | n (%) |
|---|---|
| Female | 36 (43%) |
| Age (median, IQR) | 35 (26–45) |
| Education | |
| None | 3 (4%) |
| Primary school | 37 (45%) |
| Secondary school | 19 (23%) |
| Post-secondary education/vocational | 24 (29%) |
| Occupation | |
| Informal employment | 22 (27%) |
| Formal employment | 48 (58%) |
| Not employed (including students) | 13 (16%) |
| Household size (median, IQR) | 5 (3–7) |
| Phone access (may indicate more than one) | |
| Owns a phone that no one else uses | 71 (86%) |
| Family shares a phone, primary owner | 9 (11%) |
| Family shares a phone, not primary owner | 8 (10%) |
| Other* | 1 (1%) |
| Frequency of changing SIM card in last 12 months | |
| Never | 67 (81%) |
| 1–2 times | 15 (18%) |
| Missing | 1 (1%) |
| Frequency of having airtime (calling credit) | |
| Always | 23 (28%) |
| Sometimes | 58 (70%) |
| Never | 1 (1%) |
| Missing | 1 (1%) |
| **B. Health workers surveyed (N = 22)** | **n (%)** |
| Female | 12 (55%) |
| Age (median, IQR) | 34 (28–40) |
| Years of experience working with people with TB (median, IQR) | 5 (3–8) |
| Number of people with TB treated each day (median, IQR) | 10 (5–20) |
| Number of people with TB using 99DOTS each day (median, IQR) | 4 (2–10) |
| Occupation | |
| Nurse | 15 (68%) |
| Clinical Officer | 4 (21%) |
| Counselor | 1 (4%) |
| Community Health Worker | 1 (4%) |
| Data Officer | 1 (4%) |
| Previously used a smartphone | 22 (100%) |
| Receive data from 99DOTS about people with TB | |
| Daily | 21 (95%) |
| 3–5 times per week | 1 (5%) |
| Method used to assess adherence to TB medicines^ | |
| Use 99DOTS to see if they are taking their medication | 22 (100%) |
| Ask them when they come to the health facility | 15 (68%) |
| Count the pills left when they come for a refill | 20 (91%) |
| Talk to people with TB on the phone | 17 (77%) |
| Talk to people with TB' family members | 11 (50%) |
| Home visits | 3 (14%) |
| Clinical improvement | 1 (5%) |

(*Continued*)

**Table 2.** (Continued)

| | |
|---|---|
| Experienced issues accessing 99DOTS^ | 20 (91%) |
| Application not working | 17 (77%) |
| Poor network connection | 15 (68%) |
| No electricity or power | 4 (18%) |

TB; tuberculosis; IQR: interquartile range; SIM: subscriber identity module

* This person used their community health worker's phone to call 99DOTS.

^ May indicate more than one

said that calling 99DOTS took less than one minute whereas 13 (15.7%) reported calls lasting longer than three minutes. About half of respondents reported that their health worker had shown them the 99DOTS adherence dashboard during refill visits (n = 41, 49%).

Health workers from 16 of the 18 participating health facilities were surveyed (N = 24). Two were excluded from this analysis because they reported not having access to 99DOTS data via smartphone. The 22 health workers surveyed included 15 nurses, four clinical officers, one counselor, one community health worker, and one data officer (**Table 2B**). All surveyed health workers had previous experience using a smartphone and nearly all (95%) reported receiving 99DOTS adherence data daily. All surveyed health workers reported using 99DOTS to assess adherence to TB medicines, but also confirmed dosing via pill counts, phone calls, and/or home visits.

Capability, opportunity, and motivation to use 99DOTS were assessed separately for people with TB and health workers using Likert scale-based survey questions (**S3 Table, S4 Table**).

For people with TB, the mean composite score across questions related to capability was 4.39 (95% CI 4.31–4.47). All people with TB (n = 83) agreed they knew how to use 99DOTS; however, three felt they did not get adequate training, and one other person with TB responded they did not always know which pill to take next (**Fig 2**). About 10% (n = 9) did not know where to find their health workers' contact information. Most people with TB agreed that the SMS reminders were helpful (n = 78, 94%). However, it was common for people with TB to sometimes forget to call 99DOTS after taking their TB medicines (n = 31, 37%).

The mean composite score across questions related to opportunity was 4.13 (95%CI 4.04–4.22). Stigma was a concern for about one third (n = 25, 30%) of people with TB who indicated being uncomfortable using 99DOTS in front of others or outside of the home and/or being worried 99DOTS would make it more likely others would find out they have TB. Almost all participants reported that it was easy to access a phone to make calls and did not think calling 99DOTS took too much time (n = 77, 93%). However, some struggled with phone charging (n = 39, 47%) and poor network connections (n = 40, 48%). All participants reported that 99DOTS helped make them feel more connected to their health worker and would recommend using 99DOTS to others. Most reported that they made fewer trips to the health facility because of 99DOTS (n = 73, 88%).

The mean composite score across questions related to motivation was 4.55 (95%CI 4.48–4.63). All people with TB were optimistic that 99DOTS would help them complete TB treatment and get healthy. They agreed the images and redesigned 99DOTS packaging were helpful reinforcements, and 60 (72%) reported that their health worker contacted them when they forgot to take TB medicines or call 99DOTS. Two people with TB indicated that they did not intend to call 99DOTS after taking TB medicines every day, one of whom indicated concern about the privacy of their health information. This concern was shared by 18 (22%) people with TB. However, almost all people with TB, including the one not intending to call, agreed

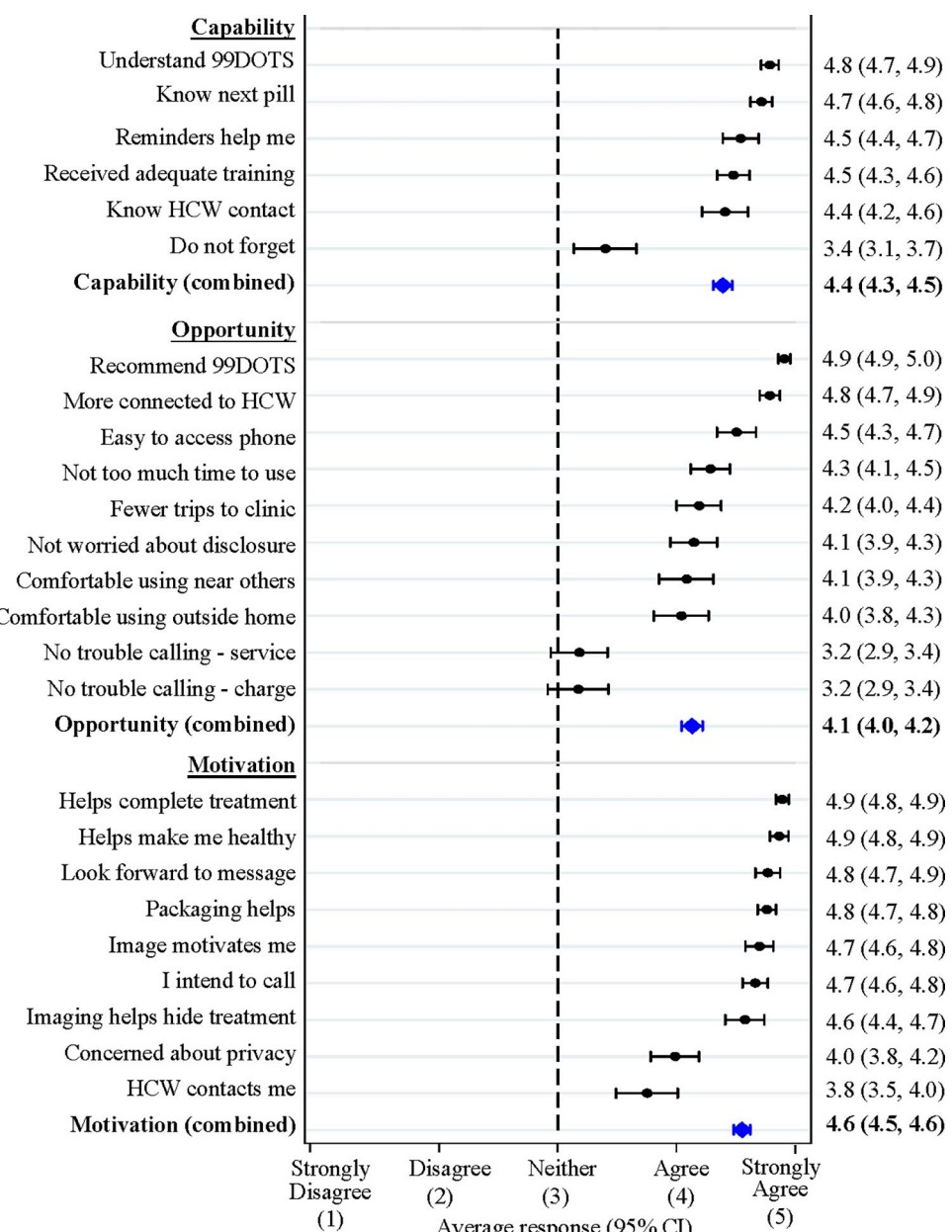

**Fig 2. Capability, Opportunity and Motivation to use 99DOTS–People with TB.** The mean and 95% CI of responses to each survey question is presented in black. Mean and 95% CI of the combined responses within each COM-B category are shown in blue. TB: tuberculosis; HCW: health care worker; COM-B: Capability Opportunity Motivation Behavior model; CI: confidence interval.

they looked forward to the motivational audio messages played when they did call 99DOTS (n = 81, 98%).

There were no significant differences in capability, opportunity, or motivation scores by gender or HIV status. There was also no significant relationship between age and capability or motivation score, but opportunity scores were significantly higher among older people with TB (correlation coefficient 0.29, p-value = 0.008) (**S1 Table, S2 Fig**).

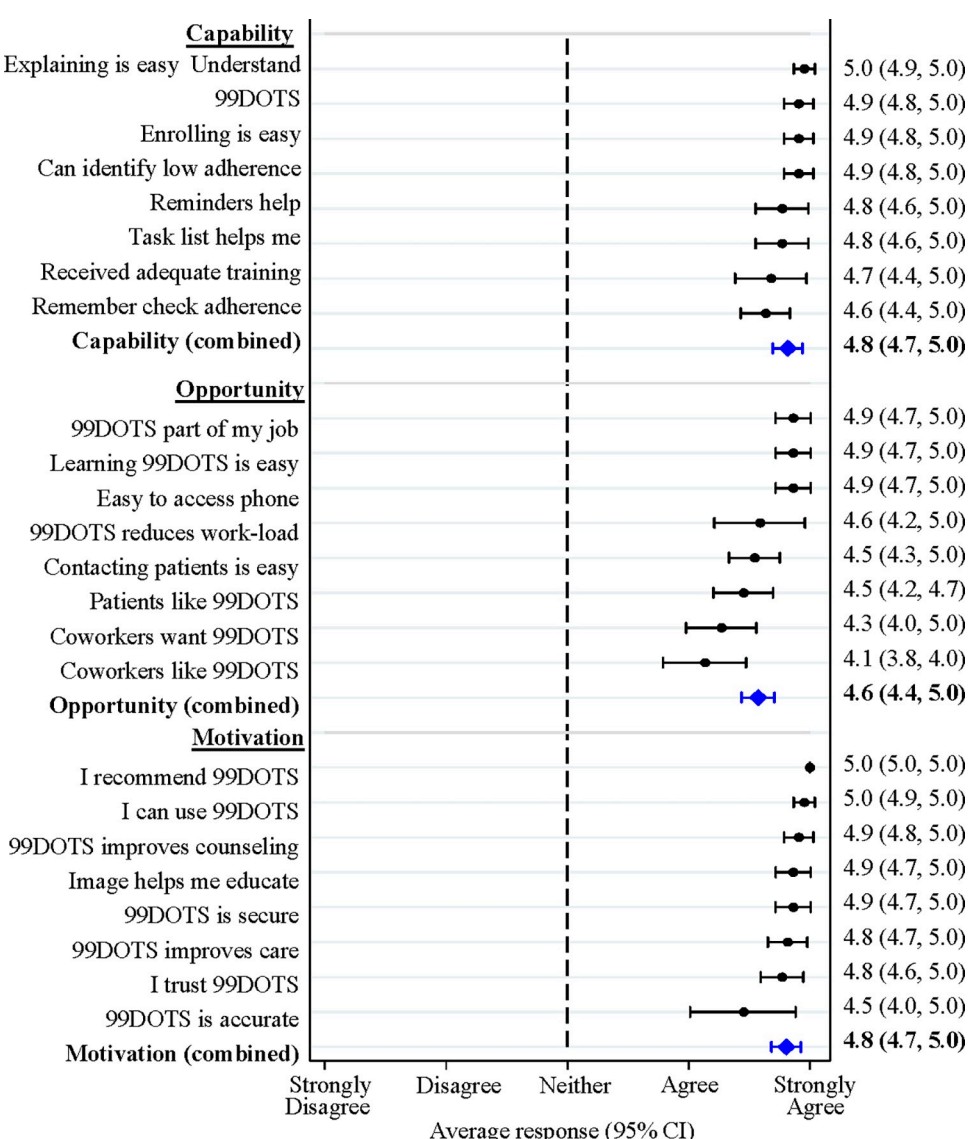

**Fig 3. Capability, Opportunity and Motivation to use 99DOTS–Health workers.** The mean and 95% CI of responses to each survey question is presented in black. Mean and 95% CI of the combined responses within each COM-B category are shown in blue. TB: tuberculosis; COM-B: Capability Opportunity Motivation Behavior model; CI: confidence interval.

For health workers, the mean composite score across questions related to capability was 4.82 (95% CI 4.69–4.94). Health workers had high scores across all capability-related questions, with only two negative responses across all 22 health workers and 8 questions (**Fig 3**). With respect to opportunity, health worker responses were even more positive than responses of people with TB with a mean composite score across questions of 4.57 (95%CI 4.44–4.71). Only two health workers responded negatively regarding coworker opinions of 99DOTS, and one did not agree that 99DOTS reduced workload. Health workers also had strong agreement across all but one question related to motivation with a mean composite score across questions of 4.81 (95%CI 4.68–4.92). Two health workers did not agree that 99DOTS accurately reflected whether people with TB took their medicines (**Fig 3**).

## Discussion

In this study nested within a highly pragmatic randomized trial conducted in Uganda, we found that 99DOTS was feasible to implement and highly acceptable to people with TB and their health workers. Over 98% of expected treatment doses were reported as taken, including over half (57.7%) by people with TB calling TFNs. Automated SMS dosing and refill reminders were delivered with high levels of success and were generally considered helpful. Both people with TB and health workers considered 99DOTS to be convenient and to increase the connection between them. Some people with TB reported stigma and privacy concerns, as well as technical issues with using 99DOTS (phone charging and poor network connection). However, our data largely represent a more favorable view of 99DOTS than presented from India, where it was reported to increased workload for health workers and reduce interaction between people with TB and health workers [3]. Acceptance of 99DOTS by people with TB in India was hindered by perceptions of reduced contact with health workers and poor counseling, and health workers reported poor facilitating conditions [5].

Engagement over time has been a key concern with 99DOTS, which requires people with TB to actively report medication dosing rather than passively recording dosing as occurs with electronic pill boxes. The 57.7% of overall doses confirmed by phone calls in our study is similar to that reported in two studies from India and a meta-analysis which included this study and 10 other DAT implementation projects [7,8]. In addition, the decreased engagement over the course of treatment is similar to that reported in India by Cross et al [2]. However, through a combination of self-report and calls to people with TB, health workers confirmed that >98% of expected doses were taken. Thomas et al also confirmed high levels of adherence (88%) among people with TB using 99DOTS in India using random urine isoniazid screening [7]. A study of 99DOTS in Tanzania that used urinalysis to confirm adherence measured by 99DOTS found a positive predictive value of 95% [9]. Although research is needed to identify opportunities to further increase engagement with 99DOTS, the real-time and quantitative dosing history provided by the platform has strong potential to enhance TB care relative to routine care.

Our study provides some of the first systematic data on perceptions of digital adherence technologies in general, and 99DOTS in particular, from sub-Saharan Africa. We found health workers and people with TB had high capability, opportunity and motivation to use 99DOTS, with composite scores between 4 and 5 for all COM-B categories indicating a response between agree or strongly agree across individual questions in each category. In surveys following implementation of 99DOTS in a mining region in Tanzania, all health workers reported that the platform helped them provide better care to people with TB, and 97.5% of people with TB felt 99DOTS helped them adhere and complete treatment. The main challenges they identified were mobile phone access, and unlike in Uganda, a minimum balance was required to call the 99DOTS platform [9]. A qualitative analysis from India reported high rates of acceptability among healthcare workers and fair acceptability among people with TB. The lower acceptability among people with TB was largely driven by technology concerns (mobile phone access, poor network signal) and stigma [5]. A minority of people with TB expressed similar concerns in our study, and the overall high scores across COM-B categories among people with TB may in part be due to our extensive customization of the 99DOTS platform using human centered design methods [10,11]. We did not find evidence that 99DOTS weakens the relationship between people with TB and health workers; all people with TB agreed that 99DOTS helps them feel more connected to their health workers, with 82% strongly agreeing. This was similar or superior to Tanzania, where 93% of people with TB agreed and 4% strongly agreed that 99DOTS helped them feel more connected to their health worker [9].

Key strengths of our study include that it was nested within a highly pragmatic trial making interpretation of its findings more broadly applicable to people with TB in Uganda and sub-Saharan Africa. Additionally, this study applied implementation science frameworks to facilitate a comprehensive and theory-based assessment of the implementation of 99DOTS. However, this study should be interpreted considering some key limitations. First, people with TB enrolled on 99DOTS may not represent the population of people with TB more broadly in Uganda. Only 52% of eligible people with TB were enrolled on 99DOTS during the parent DOT to DAT trial and there was no difference in treatment outcomes in the intention-to-treat analysis. The largest single reason for non-enrollment was lack of phone access, which is likely related to broader socioeconomic and gender differences in technology use. Finally, our survey sample was small, especially among health workers, thus limiting our ability to highlight small differences in capability, opportunity, and motivation.

In conclusion, we found that 99DOTS was feasible to implement, and more acceptable to both people with TB and providers than has been previously reported in the literature. These findings highlight the importance of adapting and contextualizing DATs prior to implementation, with specific attention to modifications that reduce stigma and enhance education/motivation as well as connection between people with TB and their health workers. Most barriers to using 99DOTS were within the opportunity domain and reflect challenges with phone charging and network connection. Further research is needed on how engagement with 99DOTS among people with TB can be enhanced and sustained throughout the treatment period, and whether the charging and network connection issues reported by a substantial proportion of people with TB can be addressed. But overall, our findings support that 99DOTS-based treatment supervision should be offered as an alternative to DOT for people with TB who have access to a phone.

## Materials and methods

### Study setting and design

This study was embedded within a stepped-wedge randomized trial that assessed the effectiveness of 99DOTS-based treatment supervision in comparison to routine care [12,13]. The trial was conducted at 18 health facilities across 15 districts of Uganda between December 2018 and January 2020 [13]. Briefly, during the intervention period at each health facility, health workers offered people with TB 99DOTS-based treatment supervision. The 99DOTS platform was contextually adapted using human centered design methods [10]. 99DOTS envelopes were re-designed to have a booklet appearance that concealed pills to reduce stigma and to include health worker contact information, pictorial pill-taking instructions, and educational/motivational messaging. The ring tone people with TB heard when making calls to report dosing was replaced with pre-recorded audio messages from local health workers expressing gratitude, encouraging medication adherence, and/or providing education around TB treatment. Other key components of the 99DOTS-based intervention included: 1) dosing reminders daily and appointment reminders prior to refill visits via short message service (SMS), 2) health worker support actions (phone calls, home visits, or additional health center visits for people with TB with sub-optimal adherence).

We conducted a longitudinal analysis to assess implementation of the key components of the 99DOTS-based intervention and cross-sectional surveys to assess feasibility and acceptability to people with TB and health workers. The study was approved by Institutional Review Boards at the University of California San Francisco and Makerere University School of Public Health. A waiver of informed consent was granted to extract demographic and clinical data

from TB treatment registers and the 99DOTS server. Survey participants provided verbal consent.

## Study population

The longitudinal assessment of TB medication adherence included all adults treated for drug-susceptible pulmonary TB and enrolled on 99DOTS in their first month of treatment during the intervention period. We excluded people with TB who transferred to other facilities to complete TB treatment.

In each health facility, surveys (**S3 Table**, **S4 Table**) were administered in months 11–13 of the trial to one or two health workers involved in TB treatment supervision and to a random sample of 10 people with TB (5 men and 5 women) who were enrolled on 99DOTS and had not already completed TB treatment.

## Data collection

**Demographic and clinical data for people with TB.**   Demographic and clinical data were obtained from the Uganda National Tuberculosis and Leprosy Programme (NTLP) treatment registers at participating health facilities. Each month, health facility staff uploaded treatment register photos to a secure, password protected server. Research staff then extracted individual-level clinical and demographic data into a secure REDCap database [14].

**TB medication adherence data.**   Enrollment on 99DOTS was confirmed using the Everwell Hub, the patient management platform available to programs using 99DOTS. The Everwell Hub includes treatment start and end dates, 99DOTS enrollment date, and total adherence (proportion of expected doses reported as taken). In addition, Everwell Health Solutions provided a detailed dosing report for each person with TB that included all days on treatment while using 99DOTS, daily dosing status (taken or missed) and method of dosing assessment (person with TB calling TFNs or health workers adding doses manually).

**SMS reminders and support actions.**   Everwell Health Solutions also provided a detailed SMS log for each enrolled person with TB that included all SMS dosing reminders sent to people with TB and their delivery status (success, invalid phone number or unsupported number type). A separate report listed support actions for each person with TB logged by health workers using the 99DOTS app.

**Acceptability surveys.**   Surveys were administered by phone in English or Luganda. All participants were reimbursed about $3 USD for their time. Surveys assessed perceptions related to using 99DOTS for TB treatment supervision [15]. Survey questions were informed by the Theoretical Domains Framework (TDF, **S3 Table**), an extension of the Capability, Opportunity and Motivation Behavior change model (COM-B). Health worker surveys also included selected constructs from the Unified Theory of Acceptance and Use of Technology (UTAUT), a consolidated framework to explain information systems usage behavior (**S4 Table**) [16,17]. The surveys included a series of statements assessing selected TDF and/or UTAUT constructs, and survey participants were asked to agree or disagree with each statement using a Likert scale (strongly disagree, disagree, neither, agree, strongly agree).

## Definitions

Adherence to TB treatment over six months of treatment while using 99DOTS was measured in two ways: 1) using only doses self-reported via calls to TFNs and 2) a combination of self-reported doses and doses confirmed manually by health workers. For the former, we counted treatment days when the person with TB called a TFN as doses taken and treatment days when no call was made as doses missed. For the latter, estimates of adherence were then augmented

to include doses confirmed manually in the 99DOTS application by health workers. People with TB who took more than 6 months of treatment due to treatment interruption or co-diagnosis with extra-pulmonary TB had adherence after 6 months censored.

Fidelity of daily SMS dosing reminders was assessed by calculating the proportion of expected SMS messages sent by the 99DOTS platform and the proportion of sent messages received by mobile phones used by people with TB. The number of expected SMS messages was calculated as the total number of days between the date of 99DOTS enrollment and the treatment end date recorded in 99DOTS. Days when people with TB called 99DOTS prior to the SMS reminder time were excluded, as the system was designed to cancel automated SMS reminders if dosing had already been confirmed on a given day.

Health worker follow-up of people with TB with poor adherence (>3 consecutive missed doses) was assessed using support actions (phone calls, home visits, or facility visits). Support actions were recorded by health workers in the 99DOTS mobile application. Implementation was measured using the number of reported support actions overall and per person with TB.

## Data analysis

We summarized fidelity to intervention components as proportions with 95% confidence intervals (CIs) or medians with inter-quartile ranges (IQRs), both overall and for subgroups. We assessed adherence per month of treatment (sequential 28-day period) from months one to six. Differences in 99DOTS engagement among people with TB (*i.e.*, proportion of expected doses reported by calling TFNs) by gender, HIV status and age quartile were compared using the non-parametric Wilcoxon rank sum test. Responses to 99DOTS acceptability surveys were scored from 1–5, with 1 being "strongly disagree" (or most unfavorable toward 99DOTS) and 5 being "strongly agree" (or most favorable). If the question was asked such that agreement indicated an unfavorable impression of 99DOTS, responses were reversed prior to analysis. Scores were summarized using means with 95% CIs. Composite scores for capability, opportunity, and motivation were created for each individual by calculating the mean response to survey questions within that category. Internal consistency among responses to survey questions related to each COM-B domain was assessed using Cronbach's alpha. Stata 15 was used for all analyses [18].

## Supporting information

**S1 Table. Adherence and Capability, Opportunity, Motivation scores by subgroups (median, IQR).**
(DOCX)

**S2 Table. Demographic and clinical characteristics of people with TB surveyed and not surveyed.**
(DOCX)

**S3 Table. Survey questions–People with TB.**
(DOCX)

**S4 Table. Survey questions–Health workers.**
(DOCX)

**S1 Fig. Engagement of people with TB with 99DOTS by gender, HIV status, and age.**
Median adherence over months 1–6 of treatment reported by phone call are shown for men and women in panel A, for participants with and without HIV in panel B, and by age quartile (18–28, 29–36, 37–48, and 49–89 years) in panel C. There was no significant difference in

trend by age (p = 0.60) or sex (p = 0.41), but people living with HIV had a significantly steeper decline in the proportion of expected doses reported by phone call overs months 1–6 of treatment, compared to people living without HIV (p<0.0001).
(DOCX)

**S2 Fig. Age and opportunity score.** Mean opportunity scores were significantly higher among older people with TB using 99DOTS (p = 0.008).
(DOCX)

**S1 Data. Data–People with TB.**
(XLSX)

**S2 Data. Data–Health workers.**
(XLSX)

**S3 Data. Data–Fidelity.**
(XLSX)

## Author Contributions

**Conceptualization:** Stavia Turyahabwe, Achilles Katamba, Adithya Cattamanchi.

**Data curation:** Alex Kityamuwesi, Maureen Lamunu, Catherine Namale, Lynn Kunihira Tinka, Agnes Sanyu Nakate, Joseph Ggita, Denis Oyuku.

**Formal analysis:** Noah Kiwanuka, Alex Kityamuwesi, Rebecca Crowder, Kevin Guzman.

**Writing – original draft:** Noah Kiwanuka, Alex Kityamuwesi, Rebecca Crowder, Kevin Guzman.

**Writing – review & editing:** Christopher A. Berger, Maureen Lamunu, Catherine Namale, Lynn Kunihira Tinka, Agnes Sanyu Nakate, Joseph Ggita, Patricia Turimumahoro, Diana Babirye, Denis Oyuku, Devika Patel, Amanda Sammann, Stavia Turyahabwe, David W. Dowdy, Achilles Katamba, Adithya Cattamanchi.

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
